# Are Subjective Intensities Indicators of Player Load and Heart Rate in Physical Education?

**DOI:** 10.3390/healthcare10030428

**Published:** 2022-02-24

**Authors:** Juan M. García-Ceberino, María G. Gamero, Sergio J. Ibáñez, Sebastián Feu

**Affiliations:** 1Optimization of Training and Sports Performance Research Group (GOERD), University of Extremadura, 10003 Cáceres, Spain; mgamerob@alumnos.unex.es (M.G.G.); sibanez@unex.es (S.J.I.); sfeu@unex.es (S.F.); 2Faculty of Humanities and Social Sciences, University of Isabel I, 09003 Burgos, Spain; 3Faculty of Education, University of Extremadura, 06006 Badajoz, Spain; 4Faculty of Sports Science, University of Extremadura, 10003 Cáceres, Spain

**Keywords:** correlation, external intensity, inertial device, internal intensity, perceived exertion, SIATE

## Abstract

Physical education teachers need valid, low-cost, subjective techniques as an alternative to high-cost new technologies to monitor students’ intensity monitoring. This study aimed to investigate the correlations between both objective and subjective external (eTL) and internal (iTL) intensities. A total of 95 primary education students participated in this study. In this regard, 40 played soccer, and 55 performed basketball tasks, recording a total of 3956 units of analysis. The intensities caused by the different soccer and basketball tasks were measured using objective techniques (inertial devices and heart rate monitors) and subjective techniques (a sheet of task analysis and ratings of perceived exertion). Matrix scatter plots were made to show the values of two variables for a dataset. In this regard, adjustment lines were plotted to determine the trend of the correlations. Then, Spearman’s correlation was calculated to measure the association between two variables. Despite the low correlation levels obtained, the main results showed significant positive correlations between the intensities. This means that the high intensity values recorded by objective techniques also implied high intensity values recorded by subjective techniques, and vice versa. Negative correlations (*r* Rho = −0.19; *p* = 0.00) were only found between the following eTL variables: task eTL per minute (subjective technique) and player load per minute (objective technique). This negative correlation occurred when students played in the same 3 vs. 3 game situation without variability in subjective eTL (M ± SD, 28.00 ± 0.00). Therefore, subjective eTL and iTL techniques could be proposed as a suitable alternative for planning and monitoring the intensities supported by students in physical education classes. Moreover, these subjective techniques are easy to use in schools.

## 1. Introduction

The planning and quantification of intensities are important for optimizing students´ physical fitness, as well as for achieving teaching objectives [1,2]. However, it is a technique rarely used by physical education teachers in primary education [3]. This may be due to poor teacher training on intensity planning and monitoring in physical education classes [4]. In this regard, teachers usually aim for a higher time of motor commitment (useful time) for the learning tasks, but they do not take into account whether the demands of these tasks are high or low. Therefore, adequate learning-task planning, using different pedagogical, organizational (time-related), and subjective intensity variables, should be performed to achieve the recommended levels of physical fitness [5,6,7].

In physical education classes, students should spend at least 50% of their time recording moderate to vigorous physical activities for adequate cardiovascular work in order to prevent diseases such as weight gain and obesity [6]. In this regard, García-Ceberino et al. [7] proposed a list of recommendations for students to spend more than 50% of the class in moderate to vigorous physical activity. Similarly, the World Health Organization [8] recommends spending at least 60 min per day in moderate to intense physical activity, mainly aerobic, throughout the week (for populations aged 5 to 17 years). In primary schools, subjective techniques can help teachers analyze whether sessions achieve recommended physical activity levels.

Intensity monitoring (this replaces the term “workload” [9,10,11]) encompasses the psychological and biological demands (internal intensity, iTL) caused by tasks or competition (external intensity, eTL). For example, the modification of game spaces or number of participants (eTL) affect the iTL of players. Thus, intensity quantification can be obtained using the eTL and iTL dimensions [12,13]. The eTL is the mechanical and locomotor stress produced by a physical activity (physical demands) [14,15]. The iTL is the physiological reaction (heart rate, HR) and stress experienced by a stimulus (eTL) [16]. HR indicates the intensity of physical activity (physiological demands) [17]. The iTL is individual and specific to each subject. A learning task, planned with the same eTL, may cause different HR values in the subjects, and these HR values will be appropriate for some subjects and inappropriate for others [18].

There are different instruments available for measuring, objectively and subjectively, the eTL and iTL. In this regard, inertial motion devices that integrate a multitude of sensors (such as accelerometers, gyroscopes, magnetometers, and GPS, among others) are used to quantify objective eTL. Among the variables recorded by these inertial devices, the most used and predictive of eTL is the Player Load (PL) [14,19]. The PL is the vectorial sum of the accelerations in three axes (vertical, anteroposterior, and lateral), and it measures the neuromuscular eTL [20]. Currently, new technologies are not accessible to all sports professionals due to their high cost [13], making it difficult to quantify these intensities. Likewise, smartphones could be integrated as an objective technique. However, many Spanish state schools prohibit their use during school periods. (This decision is made by each school.)

Faced with these problems, Ibáñez et al. [21] proposed an Integral Analysis System of Training Tasks, SIATE (Spanish acronym). The SIATE is an observational and categorization sheet, which makes it possible to quantify the subjective eTL (actions prior to sports practice). It is obtained through the sum of the categorical-ordinal values (1 to 5) assigned to six variables when categorizing learning tasks: degree of opposition, task density, percentage of simultaneous performers, competitive load, game space, and cognitive involvement. Thus, the SIATE allows the sports professionals to discover the factors that affect the sports teaching process using pedagogical, organizational, and subjective external intensity variables.

In addition, the HR (objective iTL) is measured through HR monitors that are synchronized with inertial devices [13]. The use of HR as a measure to determine the intensity of physical activity is well defined, validated, and accepted because it is a simple and non-invasive technique [12]. Its economic cost can also be high. For this reason, other types of subjective iTL instruments have appeared.

As an alternative, there is the Rating of Perceived Exertion (RPE) (psycho-physiological demands, subjective iTL). In the sports field, the most commonly used instrument is Borg’s RPE scale. The subjects indicate how tired they are by means of this scale, thus defining the intensity of the physical activity [22]. Eston and Parfitt [23] also designed a curvilinear pictorial scale (with graphic illustrations) representing the degree of perceived effort. These authors adapted Borg’s RPE scale for the child population. Therefore, subjective techniques present an alternative because of their low cost, accessibility, and ease of use [24].

The different intensities and measuring instruments mentioned above [1] are detailed in Figure 1.

After analyzing the scientific literature and instruments available to sports professionals to control and monitor sports practices, it is necessary to study their correlation because many of these instruments are not accessible to everyone. In women’s basketball training, Reina et al. [12] investigated the quantification of intensities using three instruments (WIMU^TM^ inertial devices, GARMIN^TM^ HR monitors, and the SIATE sheet) to establish correlations among them. They confirmed a direct correlation between the intensities obtained by the subjective categorization of training tasks (SIATE) with the PL and HR intensities provided by objective techniques. In soccer training, Gómez-Carmona et al. [13] also confirmed that subjective intensity variables influence objective intensity variables, finding a strong correlation between them. In addition, a combination of objective eTL and iTL (HR) factors predicted RPE in rugby training [25].

To our knowledge, the correlation between the intensities recorded by objective and subjective techniques has not been studied in the school context. Therefore, this study aimed to investigate the correlations between both objective and subjective intensities (eTL and iTL) in order to analyze the reliability of subjective techniques with respect to objective techniques. We hypothesized that correlations would be found between the following variables: (A) the objective (inertial device) and subjective (SIATE sheet) eTL variables; (B) the objective (HR) and subjective (RPE) iTL variables; and (C) the objective and subjective intensities studied.

## 2. Materials and Methods

### 2.1. Study Design

A correlational study was designed to analyze the relationships between variables in order to identify a categorical variable [26]. In turn, it was subdivided into two studies: (1) Study 1 aimed to analyze the correlations between intensities caused by different types of tasks (i.e., without opposition, individual game, small-sided games—SSGs, and full games), such that each task type implied a variability in the subjective eTL (see Table 1). (2) Study 2 aimed to analyze the correlations between intensities caused by 3 vs. 3 game situations involving the same subjective eTL (see Table 1). Study 2 also differed from Study 1 because it measured the students’ RPE. Consequently, the study was ecological and aimed to analyze the reliability of the subjective techniques with respect to the objective ones.

### 2.2. Sample

The basic unit of analysis was the record of students in each learning task type performed during nine soccer and basketball sessions (Study 1) and in two 3 vs. 3 sessions (Study 2), obtaining 3305 and 651 records, respectively. Table 1 describes the learning tasks for each study and sport. The subjective eTL of the learning tasks was always known to the teachers during the design period, i.e., before applying them in the physical education classes.

The learning tasks were played by a total of 95 primary education students from two state schools. The fifth-grade students (school 1) played soccer, while the sixth-grade students (school 2) played basketball. The soccer [27,28] and basketball [29,30] tasks are valid and reliable for application in physical education classes. The characteristics of the students are described in Table 2.

All the students who participated in at least 80% of the soccer and basketball sessions were selected as the study sample in Study 1. Of these students, those who participated in the two soccer and basketball 3 vs. 3 sessions were included in Study 2. Parents or legal guardians were required to sign an informed consent form. Similarly, the study was conducted in accordance with the ethical guidelines of the Helsinki Declaration of 1975 (with modifications in subsequent years) and Organic Law 3/2018, of December 5, on the protection of personal research data and the guarantee of digital rights (BOE, 294, 6 December 2018), to fulfil the ethical considerations of scientific research with human beings.

### 2.3. Variables and Instruments

The variables analyzed in both studies were the intensities recorded in physical education classes, grouped into eTL and iTL variables:-Objective eTL variables: (1) PL; and (2) PL per minute (PL/min). These neuromuscular eTL variables were measured using WIMU Pro^TM^ inertial devices (RealTrack System, Almería, Spain). PL was measured only during the time of motor commitment in order to eliminate distorting values, such as PL during rest periods.-Subjective eTL variables: (1) density of task, categorical–ordinal variable with five levels: 1-walking; 2-gentle pace; 3-intensity with rest; 4-intensity without rest; and 5-high intensity without rest. In Study 1, the variability of the learning tasks applied involved a different level for each task type. In contrast, the 3 vs. 3 game situations involved only one level in Study 2. (2) eTL, obtained by the sum of the values (1 to 5) given to six categorical–ordinal variables when categorizing tasks: degree of opposition, task density, percentage of simultaneous performers, competitive load, game space, and cognitive involvement. Thus, the eTL value for each learning task ranges from 6 to 30. (3) eTL*minute (eTL*min): these variables were measured through the SIATE observation sheet [21].-Objective iTL variables: (1) average HR (HR_avg_); and (2) maximum HR (HR_max_). These were measured with GARMIN^TM^ HR monitors (Garmin Ltd., Olathe, KS, USA), synchronized with the above-mentioned inertial devices through Ant+ technology [31]. HR was also measured only during the time of motor commitment.-Only in study 2 was subjective iTL (psycho-physiological demands) measured using the curvilinear pictorial scale with graphic illustrations [23], which represents the RPE.

Table 3 summarizes the variables and measurement instruments applied.

### 2.4. Procedure

First, it was necessary to obtain a series of authorizations: (1) approval from the University Bioethics Committee [Ref. 247/2019]; (2) authorization from the schools and physical education teachers; (3) approval from the school council to include the research in the curriculum of the schools; and (4) written informed consent from the parents or legal guardians.

Study 1. The students played soccer (school 1) and basketball (school 2). All learning tasks (*n* = 180) applied in both invasion sports ranged from simple (e.g., 1 vs. 0, 1 vs. 1,…) to more complex (e.g., 3 vs. 3, 4 vs. 4,…) [27,29]. These tasks were validated by experts in the field of Sport Pedagogy [28,30]. The teaching progressed based on the variability and difficulty of the tasks. The variability of the tasks implied that each one of them caused a different subjective eTL. When designing the learning tasks using the SIATE observation sheet [21], the physical education teachers were aware of their subjective eTLs before implementing them. The interventions lasted nine teaching sessions with one or two sessions per week (1 h per session). These were conducted on the school’s outdoor sports field. In this regard, the school authorities indicated the days on which the teachers could teach these invasion sports in the physical education classes. The intensity data from the soccer and basketball sessions were pooled for Study 1.

Study 2. The students played 3 vs. 3 matches with three players per team (*n* = 120), performing a five-minute warm-up. Each of the teams played against each of the other teams in this game situation. All matches were played on mini-fields. A fair balance between the teams was sought, and the teams were mixed by gender and experience. In all matches, the 3 vs. 3 game situation involved the same subjective eTL known to the teachers. A 3 vs. 3 session was performed for each sport, pre- and post-intervention (Study 1). In the soccer matches, there was no goalkeeper (with mini-goals) because their demands differ from those of field players [32]. The RPE was measured just after each match. For Study 2, the intensity data from all 3 vs. 3 matches were pooled.

In both studies, the physical education teachers (researchers) directed the soccer and basketball sessions, and then collected and analyzed the data. They had academic and professional training in these invasion sports. Inertial devices were placed using anatomical harnesses at the beginning of each session. In addition, HR monitoring with the GARMIN^TM^ device was chest-based. The data recorded from these devices were exported to the SPRO^TM^ software (RealTrack System, Almería, Spain) for quantification and statistical analysis. This methodological procedure was carried out by both researchers equally so as to guarantee the reliability and validity of the data, as suggested by Murillo-Lorente et al. [33].

Figure 2 presents the study procedure.

### 2.5. Statistical Analyses

First, the criterion assumption tests were performed to determine the use of parametric/non-parametric tests for hypothesis testing [34]. The Kolmogorov-Smirnov, Levene, and Rachas tests indicated the use of non-parametric tests.

Then, scatter plots were used to show the values of two variables for a dataset. Specifically, matrix scatter plots were used for each study. The scatter plot shows the correlation (non-causality) between the two variables. It can be positive (increase), negative (decrease) or null (the two variables are not correlated) [34]. In this regard, adjustment lines were plotted to determine the trend of the correlations.

Finally, Spearman’s correlation (Spearman’s Rho) for non-parametric tests was performed. It measures association between two variables and values from -1 to +1, indicating negative or positive associations, respectively. A positive association indicates that high values of one variable are correlated with high values of the other variable, and vice versa. On the other hand, a negative association indicates that high values of one variable are correlated with low values of the other variable, and vice versa [34]. The following interpretation scale determines the level of association: null = 0.00; between null and low = 0.00–0.25 (−0.25); low = 0.26 (−0.26)–0.50 (−0.50); between moderate and strong = 0.51 (−0.51)–0.75 (−0.75); and, between strong and perfect = 0.76 (−0.76)–1.00 (−1.00) [35].

The same scale was always used for each instrument. Therefore, Spearman’s Rho was not affected by changes in the units of measurement [35].

Graphics and statistical analysis were performed with SPSS 25.0 (Released 2017. IBM SPSS Statistics for Windows, Version 25. IBM Corp., Armonk, NY, USA).

## 3. Results

Study 1. Figure 3 shows the matrix scatter plots between the intensities studied (in pairs). These indicate that the possible correlations follow a positive/increase trend.

Table 4 shows the correlation results between the objective eTL, the subjective eTL, and the objective iTL variables. The objective intensities (eTL and iTL) were recorded during the application of the intervention programs. In contrast, the design of the learning tasks prior to their application implies a variability of subjective eTL (*M* ± *SD*, 17.15 ± 4.93, minimum 8–maximum 30). The main results show significant positive correlations between all the intensity variables studied, both objective and subjective. Therefore, high values of the subjective eTL variables also indicate high values of the objective eTL and iTL variables, and vice versa. Moreover, the moderate to strong correlation between the PL/min variable (eTL) and the HR_avg_ and HR_max_ variables (iTL) is noteworthy. This level of correlation is similar to that between PL and HR_max_ variables.

Study 2. The matrix scatter plots between the intensities studied (in pairs) are shown in Figure 4. These indicate that the possible correlations follow a positive/increase trend, except for the variables: eTL*min and PL/min (negative correlation).

Table 5 shows the correlation analysis between all of the objective and subjective intensities studied. The 3 vs. 3 session design prior to the application implies the same subjective eTL (*M* ± *SD*, 28.00 ± 0.00). Although the correlation coefficient associated with the RPE is low, the most salient results indicate significant positive correlations between the RPE and objective variables of eTL (PL and PL/min) and iTL (HR_avg_). Thus, high values of the RPE also indicate high values of the objective eTL and iTL variables mentioned, and vice versa. PL continues to show moderate to strong correlation with HR.

Finally, Table 6 shows an equivalence of the iTL variables compared to Buceta’s values [36]. Thus, students should be trained before using the RPE pictorial scale [23] because they tend to show lower values than the real ones.

## 4. Discussion

In sports practices, more accessible and manageable devices would be necessary, to which not all teachers have access. The study aimed to investigate the correlations between both objective and subjective intensities (eTL and iTL) to analyze the reliability of subjective techniques with respect to objective techniques. For this purpose, it was necessary to quantify the intensities caused by different soccer and basketball tasks in primary education (5th and 6th grades). In general, despite the low levels obtained, the main findings indicated that the significant correlations recorded between the intensities were mostly positive/increasing. Thus, high intensity values recorded by objective techniques also mean high intensity values recorded by subjective techniques, and vice versa.

Study 1 (intensities refer to intervention programs). There were significant positive correlations, with an increasing trend, between the eTL variables recorded using objective and subjective techniques. In the sports training context, Gómez-Carmona et al. [13] found a very strong correlation between the objective (WIMU Pro^TM^ devices) and subjective (SIATE observation sheet) eTL variables when monitoring U–19 soccer players. Also, Reina et al. [12] indicated that there was a correlation between the values obtained by both eTL measurement techniques when monitoring players from a women’s senior basketball team. Therefore, it could be stated that the subjective eTL variables, used when designing the learning tasks, influence the objective eTL variables recorded when these tasks are later applied in sports play.

Likewise, the objective iTL (HR) variables correlated positively with all eTL variables, both objective and subjective. In a previous correlational study [12], a direct correlation was confirmed between the intensities recorded by subjective categorization of basketball tasks and the PL and HR recorded using objective techniques.

Movement intensity and HR analysis are traditional techniques used by sports professionals to measure the physical [14] and physiological [17] demands in invasion sports, respectively. Thus, the SIATE observation sheet [21] is proposed as a subjective technique for monitoring eTL without the need for employing expensive and complex technologies to use. In turn, this subjective eTL technique is known to correlate with HR [12,13], and it has been employed by different researchers for the planning of soccer [5,27,37] and basketball [38,39,40] sessions in physical education. The highest correlations (moderate to strong) were recorded between the PL and HR variables. Thus, the higher the accelerometry, the higher the task intensity, and vice versa.

In conclusion, when learning tasks causing different subjective eTLs were employed, the hypotheses A and C were partially fulfilled (Table 4).

Study 2 (intensities refer to 3 vs. 3 sessions). The RPE (psycho-physiological demand) can provide an adequate estimate of real HR, allowing us to know the intensity during sports play [22]. In terms of iTL, despite the low values obtained, the main results indicated significant positive correlations between the RPE and HR_avg_. In this regard, previous studies in the context of sports training have established a high correlation between both measurement techniques, and they indicate the use of RPE as a valid method to monitor exercise in the absence of other costly and sophisticated tools [41,42], e.g., HR monitors. Alexiou and Coutts [43] suggested that the RPE correlates better with the HR in less intermittent sessions working on aerobic endurance when analyzing elite women’s soccer players. This may be because HR_max_ and RPE are lower in aerobic sessions [33]. In this regard, the type of task (teaching means) has a direct effect on kinematic, neuromuscular, and internal demands [7,13] and, together with the measurement technique used, can condition the magnitude and uncertainty of the associations [44].

Furthermore, the RPE correlated positively with the objective variables of PL and PL/min. In line with the results described above, Lovell et al. [25] stated that a combination of objective eTL and iTL (HR) factors predicted the RPE in professional rugby players. Likewise, McLaren et al. [44] stated that RPE and HR showed positive associations with eTL derived from running and accelerometry in team sports. PL continues to show moderate to strong correlation with HR. Therefore, in this study, there was an increasing trend in the values recorded through objective and subjective techniques. It is necessary to previously train students in the use of the RPE pictorial scale [23] in physical education. This subjective iTL technique is very popular among sports professionals due to its easy accessibility, simplicity, and accuracy [45]. Despite this, the data obtained can be compromised by the interaction of multiple factors [24], e.g., the psychological perspective [33], or previous experience in its use. In addition, Borresen and Lambert [41] observed that the data are not as accurate when proportionally more time is spent training at low or high intensity, compared to HR.

Another relevant result was the negative correlation between eTL*min and PL/min variables. Thus, high values of eTL*min indicated low values of PL/min, and vice versa. These results are in line with those obtained in previous studies on school soccer [1,7]. The tasks involving playing situations, with the simultaneous participation of the players and in reduced game-spaces, invoke a medium-high subjective eTL [27] and high intensity [7], while the PL recorded is low because these tasks cause few accelerations as players cover longer distances at more constant speeds [46,47]. However, the tasks without opposition and with consecutive participation (organization in rows) of the players involve a medium-low subjective eTL [27], while the accelerations and PL recorded are higher than those obtained in playing situations [1]. This type of task is performed repeatedly and with short breaks. The players move a few meters, accelerating to the maximum from the beginning of the tasks to decelerating quickly [48]. For this reason, it is important to pay attention to the design of the learning tasks used in physical education because it subsequently influences the physical and physiological demands supported by the students.

In conclusion, when the same learning tasks involving equal subjective eTL were employed, hypotheses B and C were partially fulfilled (Table 5).

## 5. Limitations and Practical Applications

A study limitation was the sample size, as well as not having trained the students in the use of the RPE scale. Although no strong correlations were recorded, significant and positive correlations were found, indicating that the intensity values increased or decreased equally throughout the soccer and basketball sessions.

Subjective techniques (SIATE observation sheet and RPE scale) are a suitable alternative because they help teachers to know the impact of learning tasks on students and to improve their physical fitness, since high objective values imply high subjective values, and vice versa. In addition, subjective techniques can easily be used in physical education, and they have low economic costs. However, further studies are needed to confirm the homogeneity of the objective and subjective measures.

## 6. Conclusions

The design of the learning tasks and knowledge of their subjective external intensities provide feedback on the intensities that will subsequently occur when applying them: the neuromuscular and kinematic intensity variables, the heart rate, and the ratings of perceived exertion. In this way, the proposed fitness physical and health objectives indicated in the Spanish curriculum and scientific literature could be met, i.e., ≥50% of class time students should spend in moderate to vigorous physical activity for adequate cardiovascular work and health improvement.

This study could confirm the usefulness of the SIATE observation sheet and RPE as subjective techniques for the prediction of objective intensities in physical education classes since subjective measures provide a significant (not strong) correlation when compared to an objective standard. Also, these subjective techniques can easily be used by teachers. Knowledge of subjective measures will allow teachers to optimize the teaching–learning process. It is recommended for further studies that students have previous experience in the use of the RPE scale in school. Furthermore, teachers must be trained in the use of the SIATE observation sheet.

## Figures and Tables

**Figure 1 healthcare-10-00428-f001:**
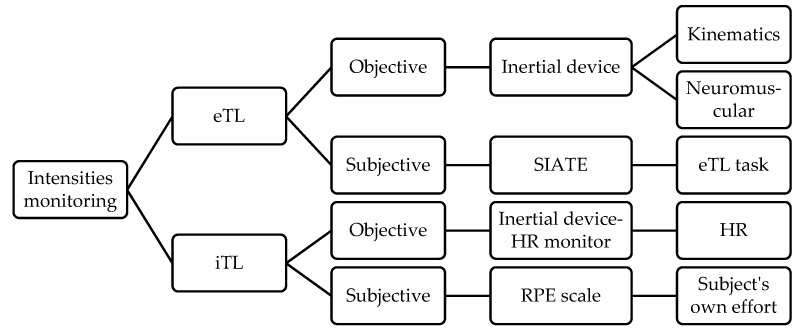
Classification of the intensities and measuring instruments according to García-Ceberino et al. [1]. Note: eTL = External Intensity; SIATE = Integral Analysis System of Training Tasks; iTL = Internal Intensity; HR = Heart Rate; RPE = Rating of Perceived Exertion.

**Figure 2 healthcare-10-00428-f002:**
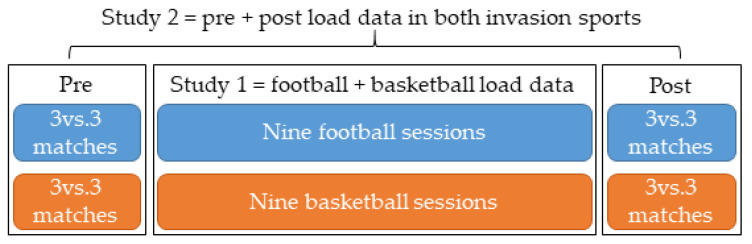
Procedure for both studies.

**Figure 3 healthcare-10-00428-f003:**
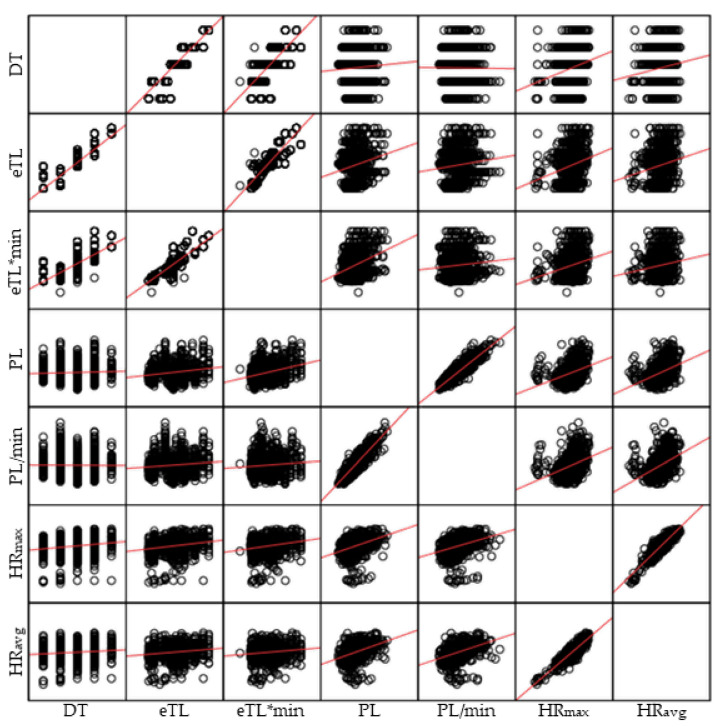
Matrix scatter plots between the intensities studied (in pairs) in the interventions. Note: eTL = External Intensity; TD = Task Density; PL = Player Load; min = Minute; iTL = Internal Intensity; HR = Heart Rate; avg = Average; max = Maximum.

**Figure 4 healthcare-10-00428-f004:**
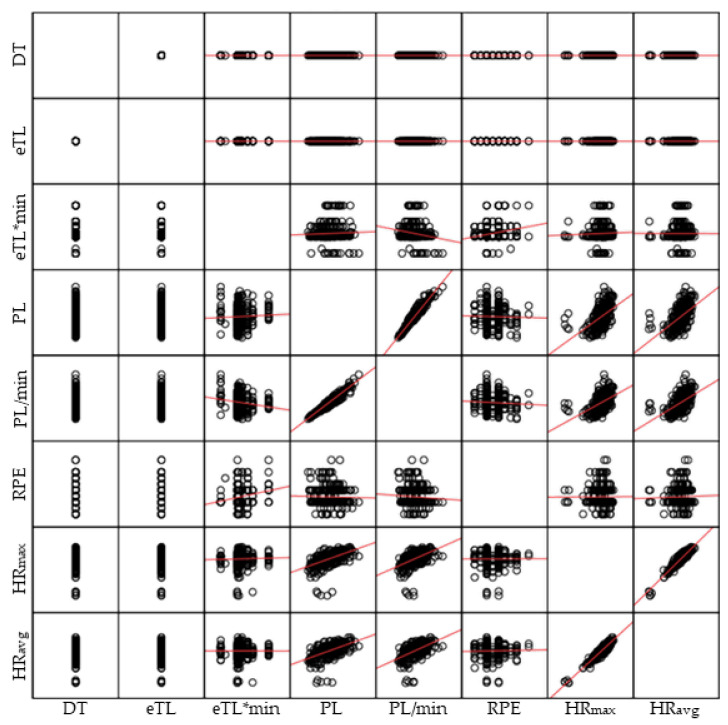
Matrix scatter plots between the intensities studied (in pairs) in the 3 vs. 3 matches. Note: eTL = External Intensity; TD = Task Density; PL = Player Load; min = Minute; iTL = Internal Intensity; HR = Heart Rate; avg = Average; max = Maximum.

**Table 1 healthcare-10-00428-t001:** Description of the learning tasks (by study and sport).

Study	Task Type	Example	Soccer		Basketball	
			%	eTL_avg_	%	eTL_avg_
Study 1 Tasks with different subjective eTL	Without Opposition	1 vs. 0…	39.90	10.85	56.50	12.03
	Individual Game	1 vs. 1	28.30	17.83	15.70	19.52
	Inequality SSG	2 vs. 1…	25.10	20.84	17.80	19.17
	Equality SSG	2 vs. 2…	1.80	17.00	5.70	21.48
	Full Game	5 vs. 5	4.90	28.64	4.30	24.66
Study 2 3 vs. 3 with same subjective eTL	Equality SSG	3 vs. 3	100.00	28	100.00	28.00

Note: eTL = External Intensity; avg = Average; SSG = Small-Sided Game.

**Table 2 healthcare-10-00428-t002:** Characteristics of the students participating (by study).

Demographic Data	Study 1 Tasks with Different Subjective eTL		Study 2 3 vs. 3 with Same Subjective eTL	
School, grade	School 1, 5th PE	School 2, 6th PE	School 1, 5th PE	School 2, 6th PE
Students, girls	40, 18 girls	55, 32 girls	33, 16 girls	48, 25 girls
Years (M ± SD)	10.65 ± 0.48	11.09 ± 0.29	10.67 ± 0.48	11.10 ± 0.31

Note: M = Mean; SD = Standard Deviation; eTL = External Intensity; PE = Primary Education.

**Table 3 healthcare-10-00428-t003:** Summary of study variables and instruments.

Intensity	Variable	Unit	Description	Instrument
eTL (objective)	PL	Arbitrary units (per min)	Neuromuscular eTL resulting from accelerations	WIMU Pro^TM^
	PL/min			
eTL (subjective)	Task density	Scale 1 to 5	Intensity of the learning task	SIATE observation sheet
	Task eTL	Number 6 to 30 (per min)	Intensity resulting from the sum of six categorical variables	
	eTL*min			
iTL (objective)	HR_avg_	Beats per minute	Number (average/maximum) of beats per minute	GARMIN^TM^ monitors
	HR_max_			
iTL (subjective)	RPE	Scale 1 to 10	Perception of one’s own effort	CPS (graphics)

Note: eTL = External Intensity; PL = Player Load; min = Minute; SIATE = Integral Analysis System of Training Tasks; iTL = Internal Intensity; HR = Heart Rate; avg = Average; max = Maximum; RPE = Ratings of Perceived Exertion; CPS = Curvilinear Pictorial Scale.

**Table 4 healthcare-10-00428-t004:** Correlation analysis between the intensities recorded when applying the programs.

Intensity	Spearman’s Rho	Objective iTL		Objective eTL		Subjective eTL	
		HR_avg_	HR_max_	PL/min	PL	eTL*min	eTL
TD	*r*	0.26 **	0.31 **	0.23 **	0.30 **	0.78 **	0.87 **
	*p*	0.00	0.00	0.00	0.00	0.00	0.00
	*n*	3205	3175	3273	3273	1088	1480
eTL	*r*	0.26 **	0.33 **	0.24 **	0.36 **	0.89 **	
	*p*	0.00	0.00	0.00	0.00	0.00	
	*n*	3205	3175	3273	3273	1088	
eTL*min	*r*	0.12 **	0.22 **	0.09 *	0.27 **		
	*p*	0.00	0.00	0.02	0.00		
	*n*	1031	1031	1088	1088		
PL	*r*	0.47 **	0.53 **	0.66 **			
	*p*	0.00	0.00	0.00			
	*n*	3175	3175	3273			
PL/min	*r*	0.57 **	0.51 **				
	*p*	0.00	0.00				
	*n*	3175	3175				
HR_max_	*r*	0.88 **					
	*p*	0.00					
	*n*	3175					

Note: *n* = Cases Analyzed; *r* = Spearman’s Correlation; eTL = External Intensity; TD = Task Density; PL = Player Load; min = Minute; iTL = Internal Intensity; HR = Heart Rate; avg = Average; max = Maximum; ** Significant correlation at the 0.01 level; * Significant correlation at the 0.05 level.

**Table 5 healthcare-10-00428-t005:** Correlation analysis between the intensities recorded when apply to the 3 vs. 3 matches.

Intensity	Spearman’s Rho	Ob. iTL		Sub. iTL	Ob. eTL		Sub. eTL	
		HR_avg_	HR_max_	RPE	PL/min	PL	eTL/min	eTL
TD	*r*	0.15 **	0.11 **	-	0.04	0.06	0.60 **	1.0 **
	*p*	0.00	0.00	-	0.27	0.09	0.00	-
	*n*	711	711	674	722	722	400	728
eTL	*r*	0.15 **	0.11 **	-	0.04	0.06	0.60 **	
	*p*	0.00	0.00	-	0.27	0.09	0.00	
	*n*	711	711	674	722	722	400	
eTL*min	*r*	0.12 *	0.13 **	0.08	−0.19 **	−0.08		
	*p*	0.02	0.01	0.15	0.00	0.13		
	*n*	390	390	346	400	400		
PL	*r*	0.61 **	0.54 **	0.21 **	0.94 **			
	*p*	0.00	0.00	0.00	0.00			
	*n*	805	805	668	817			
PL/min	*r*	0.57 **	0.53 **	0.15 **				
	*p*	0.00	0.00	0.00				
	*n*	805	805	668				
RPE	*r*	0.14 **	0.08					
	*p*	0.00	0.05					
	*n*	658	658					
HR_max_	*r*	0.91 **						
	*p*	0.00						
	*n*	805						

Note: *n* = Cases Analyzed; *r* = Spearman’s Correlation; Ob. = Objective; Sub. = Subjective; eTL = External Intensity; TD = Task Density; PL = Player Load; min = Minute; iTL = Internal Intensity; HR = Heart Rate; avg = Average; max = Maximum; RPE = Ratings of Perceived Exertion; ** Significant correlation at the 0.01 level; * Significant correlation at the 0.05 level.

**Table 6 healthcare-10-00428-t006:** Equivalence of the iTL variables compared to Buceta’s values [36].

HR_avg_	HR_max_ M(SD)	RPE	Borg Scale Equivalence	Approximate in bpm	Degree of Stress Intensity (% of max. Capacity)
170.51 (22.18)	188.33 (21.37)	3.40 (1.86)	Fairly light	110–130	30

Note: M = Mean; SD = Standard Deviation; iTL = Internal Intensity; HR = Heart Rate; avg = Average; max = Maximum; RPE = Ratings of Perceived Exertion; bpm = Beats per Minute.

## Data Availability

Data will be available upon reasonable request to the corresponding author.

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
