# Peer review of "Are Subjective Intensities Indicators of Player Load and Heart Rate in Physical Education?"

_healthcare, 2022, doi:10.3390/healthcare10030428_

Round 1
Reviewer 1 Report
The manuscript has improved. However there are some issues:
- A complete review of English spelling and expressions is urgently needed (i.e. Line 380) "....for teachers to use....".
- Originality of the topic. RPE and SIATE are well described tools. There are several investigations about their correlation with internal load.
Author Response
Dear reviewer,
We would like to express our gratitude again to reviewer 1 for the time in reviewing our manuscript and for providing us comments helpful to improve this manuscript quality. We have answered their concerns (all corrections were marked in red).
--------------------
Reviewer’ note: A complete review of English spelling and expressions is urgently needed (i.e. Line 380) "...for teachers to use...".
Authors’ response: Thank for your suggestion. A qualified native translator performed a grammatical revision of the manuscript (certificate attached).
--------------------
Reviewer’ note: Originality of the topic. RPE and SIATE are well described tools. There are several investigations about their correlation with internal load.
Authors’ response: Thank you for your comments. The introduction describes several studies that analyze the correlation between objective and subjective techniques (lines 103 to 111).
Gómez-Carmona, C.D.; Gamonales, J.M.; Feu, S.; Ibáñez, S.J. Estudio de la carga interna y externa a través de diferentes instrumentos. Un estudio de casos en fútbol formativo. Sportis Sci J 2019, 5, 444-468.
Lovell, T.W.J.; Sirotic, A.C.; Impellizzeri, F.M.; Coutts, A.J. Factors Affecting Perception of Effort (Session Rating of Perceived Exertion) During Rugby League Training. International Journal of Sports Physiology and Performance 2013, 8, 62-69.
Reina, M.; Mancha‐Triguero, D.; García‐Santos, D.; García‐Rubio, J.; Ibáñez, S.J. Comparación de tres métodos de cuantificación de la carga de entrenamiento en baloncesto. RICYDE. Revista Internacional de Ciencias del Deporte 2019, 15, 368-382.

Reviewer 2 Report
I thank the authors for the responses to my comments and although the article has improved considerably, I consider making some slight improvements. Nevertheless, I am very happy to be able to revise this article again with all the improvements.
- The importance of quantifying loads in Physical Education has been indicated. Thank you.
- The sample size has been included as a limitation. Thank you.
- The recommendation about the reasons why Physical Education teachers do not usually quantify and control loads has been incorporated (lines 38-42). However I consider that it still does not answer my question in a solid way and a better argumentation of this section is needed.
The rest of the comments have also been correctly answered and justified.
I consider that the article is much improved and could be published.
Author Response
Dear reviewer,
We would like to express our gratitude again to reviewer 2 for the time in reviewing our manuscript and for providing us comments helpful to improve this manuscript quality. We have answered their concerns (all corrections were marked in red).
--------------------
Reviewer’ note: The recommendation about the reasons why Physical Education teachers do not usually quantify and control loads has been incorporated (lines 38-42). However, I consider that it still does not answer my question in a solid way and a better argumentation of this section is needed.
Authors’ response: Thank you for your suggestion. It was indicated that physical education teachers need more training in workloads’ planning and monitoring (lines 37 to 39).
Sierra, R. Formación docente para el control de la carga en la clase de educación física. Revista de Investigación en Educación 2005, 2, 33-48.
We would also like to thank you for considering this manuscript suitable for publication.
Reviewer 3 Report
Thank you for giving me the opportunity to review your work. I believe that creation and validation of subjective methods of PA intensity assessment are needed on every level, particularly in children. In primary schools it can help teachers to analyze whether training plans are adequate to recommended weekly PA level in that group (on average at least 1 hour of vigorous PA activity daily as per WHO 2020 guidelines). I have some comments:
- Please include the information on the recommended PA levels in children in the introduction section and a short information on how used PA settings fitted into the recommendations. It would be an additional important information regarding the necessity to assess PA intensity in schools.
- Please explain all abbreviations at first mention.
- All of the observed correlations, except derivative variables are rather weak. In my opinion the conclusion should be softened or even opposite. The study shows that it is really hard to subjectively analyze the PA intensity in this group of children and therefore other measures are required.
- Please add information on the type of HR monitoring with Garmin device - was it chest-based or wrist-based?
- Moderate English editing is still needed - for example Line 130 "were including"
Author Response
Dear reviewer,
We would like to express our gratitude again to reviewer 3 for the time in reviewing our manuscript and for providing us comments helpful to improve this manuscript quality. We have answered their concerns (all corrections were marked in red).
--------------------
Reviewer’ note: Please include the information on the recommended PA levels in children in the introduction section and a short information on how used PA settings fitted into the recommendations. It would be an additional important information regarding the necessity to assess PA intensity in schools.
Authors’ response: Thank you for your suggestion. Information on recommended physical activity levels in physical education classes has been added (lines 44 to 52).
Aznar, S.; Webster, T. Actividad Física y Salud en la Infancia y la Adolescencia. Guía para todas las personas que participan en su educación; Ministerio de Educacion y Cultura, Centro de Investigación y Documentación educativa: Madrid, Spain, 2009.
García-Ceberino, J.M.; Feu, S.; Antúnez, A.; Ibáñez, S. Organization of Students and Total Task Time: External and Internal Load Recorded during Motor Activity. Applied Sciences 2021, 11, 10940.
World Health Organization. Physical activity. Availabe online: https://www.who.int/news-room/fact-sheets/detail/physical-activity (accessed on 15/02/2022).
--------------------
Reviewer’ note: Please explain all abbreviations at first mention.
Authors’ response: We agree with you. The meaning of all abbreviations has been indicated the first time they appear in the manuscript. Abbreviations are also indicated in the figures and tables to assist the reader in reading them.
--------------------
Reviewer’ note: Please add information on the type of HR monitoring with Garmin device - was it chest-based or wrist-based?
Authors’ response: Based on your question, it has been indicated that HR monitoring with the GARMIN device was chest-based (lines 212 to 213).
--------------------
Reviewer’ note: All of the observed correlations, except derivative variables are rather weak. In my opinion the conclusion should be softened or even opposite. The study shows that it is really hard to subjectively analyze the PA intensity in this group of children and therefore other measures are required.
Authors’ response: Thank you for your comment. The failure to obtain strong correlations and the need to train students in the use of the RPE scale have been added in the limitations section (lines 380 to 384). The conclusions have also been softened. In addition, it has been proposed to continue with the comparison of objective and subjective techniques in the school (prospective research) (lines 389 to 390).
--------------------
Reviewer’ note: Moderate English editing is still needed - for example Line 130 "were including".
Authors’ response: Thank for your suggestion. A qualified native translator performed a grammatical revision of the manuscript (certificate attached).

Round 2
Reviewer 3 Report
Thank you for taking time to make suggested changes to the text. I have no further comments.
Author Response
Dear reviewer,
Thank you for considering our manuscript for publication.
Kind regards.
This manuscript is a resubmission of an earlier submission. The following is a list of the peer review reports and author responses from that submission.
Round 1
Reviewer 1 Report
The methods and the design of the study are just OK but there are some important issues:
- The significance of the contents is poor.
- RPE and SIATE are well described and validated tools for load assessment. What is new in the article?
- Huge English mistakes (such as page 139 or page 147)
- The parameters for Ext and Int Load are at different scales
- The assumption that the study confirms the usefulness of the SIATE observation sheet and RPE as subjective techniques for the prediction of objective loads in physical education classes is very risky. Not enough evidences can be observed in the results.
- Perhaps an approach with training participants in the use of RPE and SIATE would have more sense
- Study 1 tasks are poorly described. Examples are not enough
- Practical applications: add some more
- Page 163. Rephrase last sentence
Reviewer 2 Report
Please refer to the attached file.

Reviewer 3 Report
The objective of this work is to study the correlations between objective and subjective external and internal loads. I consider that it is a work that takes care of the formal aspects of a scientific article and I enjoyed reading this article that adds some important research to the field of Physical Education.
The real strengths of the paper are included in the methodology section which is clear and transparent. The statistical analysis is clearly defined.
However, some suggestions for improving the article are detailed below.
Firstly, I consider that the sample studied is not sufficiently representative to be able to draw such solid conclusions as the authors suggest, so it should be included in limitations.
I consider that this study would have been more pertinent if it had been carried out in secondary education. I have the doubt that the authors state in line 333 that the proposed fitness and health objectives indicated in the Spanish curriculum can be met, but they do not define them.
Line 35. I would appreciate it if the answer to the following question could be expanded in a reasoned way: why is it important to quantify the workloads in Physical Education in Primary School?
Line 37. I would appreciate an argumentation of the reasons why Physical Education teachers do not usually quantify and control the loads.
I look forward to rereading this article after the suggested improvements.